# Open-Source 3D Printing in the Prosthetic Field—The Case of Upper Limb Prostheses: A Review

Kevin Wendo [1,2,*], Olivier Barbier [1,3], Xavier Bollen [4], Thomas Schubert [1,3], Thierry Lejeune [1,5], Benoit Raucent [4] and Raphael Olszewski [1,2,6]

1 Neuro Musculo Skeletal Lab (NMSK), Institut de Recherche Expérimentale et Clinique (IREC), UCLouvain, Université Catholique de Louvain, 1200 Brussels, Belgium; olivier.barbier@saintluc.uclouvain.be (O.B.); thomas.schubert@uclouvain.be (T.S.); thierry.lejeune@saintluc.uclouvain.be (T.L.); raphael.olszewski@saintluc.uclouvain.be (R.O.)
2 Oral and Maxillofacial Surgery Lab (OMFS Lab), NMSK, IREC, UCLouvain, Université Catholique de Louvain, 1200 Brussels, Belgium
3 Department of Orthopedic Surgery, Cliniques Universitaires Saint-Luc, 1200 Brussels, Belgium
4 Institute of Mechanics, Materials and Civil Engineering, UCLouvain, Université Catholique de Louvain, 1348 Louvain-La-Neuve, Belgium; xavier.bollen@uclouvain.be (X.B.); benoit.raucent@uclouvain.be (B.R.)
5 Physical Medicine and Rehabilitation Department, Cliniques Universitaires Saint-Luc, 1200 Brussels, Belgium
6 Department of Oral and Maxillofacial Surgery, Cliniques Universitaires Saint-Luc, 1200 Brussels, Belgium
* Correspondence: kevin.wendo@uclouvain.be

**Abstract:** Upper limb loss alters individuals' private and professional life. Prosthetic devices are thus a solution to supply the missing upper limb segments. Nevertheless, commercial prostheses are often unaffordable, or inaccessible, to underprivileged individuals (e.g., no health insurance, low incomes, warzone). Among potential affordable alternatives, additive manufacturing, commonly "3D printing", has been increasingly employed. This technology offers higher availability and accessibility, and can produce complex geometrical and highly customized products, which are essential features for prostheses manufacturing. Therefore, this study aims to portray an overview of reliable open-source upper limb 3D-printed prostheses currently available. We thus searched the scientific literature and online repositories hosting 3D-printable designs. We extracted data relative to mechanical and kinematic properties, 3D printing process and efficacy for each device. We found six studies implementing open-source 3DP upper limb prostheses and twenty-five open-source designs from online databases meeting selection criteria. Devices' technical specifications were not systematically reported. In conclusion, though open-source 3D-printed upper limb prostheses can perform some functional tasks and grasps, and are widely employed to supply limb differences, further research is mandatory to validate their usage and to prove their clinical efficacy. More guidelines are required to unify contributions from private makers and non-governmental organizations with scientific groups.

**Keywords:** prosthetic; prosthesis; prostheses; upper limb; hand; arm; open-source; 3D printing; 3D model

## 1. Introduction

Upper limb difference, due to congenital defect or trauma, is a devastating condition for patients, either young or old. Indeed, such impairment alters their self-esteem and social life as well as their abilities to perform daily activities [1,2]. Therefore, consequences of this type of disability are both physical and psychological [1–3]. For children, limb loss can also lead to perturbations of their psychomotor development [3]. Unfortunately, numbers of individuals suffering from that condition increase every year and are more prevalent in some areas (e.g., warzones, low-income countries, etc.) [4]. For example, in 2005, 541,000 US citizens suffered from upper limb loss and this amount is expected to double by 2050 [5]. In 2010, The Centers for Disease Control and Prevention estimated approximately that 4 out of 10,000 live births were newborns with upper limb reductions [6].

These estimations are likely to increase due to the world population constant growth [7]. In 2017, The World Organization of Health (WHO) reported that around 0.5% of the world population were suffering from limb loss or differences and needed an assistive device [8]. This number represents millions of individuals which the international organization expects to constantly increase as the world population continues to grow and life expectancy to extend [8]. Furthermore, WHO also estimated that only 1 in 10 individuals in need of assistive devices (prothesis or orthosis) had access to them, due to their high cost, their poor availability, the absence of trained personnel and the lack of institutional structure [8]. Moreover, most affected individuals reside in countries where their access to appropriate healthcare or their personal resources, often both, are limited [4].

Solutions to address upper limbs loss, or absence, of function, and to limit their associated psychological impact have existed since antiquity, in the form of prostheses [9]. Indeed, prosthetic devices were fitted onto traumatically amputated soldiers from the battlefield [9]. Since then, prostheses, initially made for cosmetic purposes, were continuously revised and improved. Today, commercial upper limb prosthetic solutions (hands, forearms, arms) are more anthropomorphic, able to perform various movements and grips, and some attempt to restore sensitivity to the affected limb [2,9]. Nevertheless, the degree of personalization according to the final user's anatomy and aesthetic taste can be limited. Although these technological advancements in commercial prostheses can be very beneficial for some individuals, for others their cost remains a limit [3,4,10]. This aspect is more pronounced among the pediatric population as their growth requires periodic changes of prosthetic devices [3,11].

In order to supply the increasing demand in underprivileged environment, diverse initiatives, both from individuals and associations, took place. They implemented 3D (three-dimensional) printing technology in their actions in order to offer a 3D-printed upper limb prosthesis to people manifesting their need [11–13].

3D printing is an expanding technology consisting of producing a physical 3D object from its 3D digital model. The first 3D-printed upper limb prosthesis was produced in 2012, a hand prosthesis baptized "Robohand" [11]. Since then, the technology continued to evolve and to mature with new other 3D-printed prosthetic devices made available, but in some aspects, it is still in its infancy [3,11–13]. Indeed, additional developments are required to refine the anthropomorphism and cosmesis of 3DP prostheses, and to improve their comfort, robustness and functionality [3,11–13]. Hand prosthetics are the most designed and printed ones. Moreover, most 3D-printed upper limb prostheses are intended for children [11]. This trend is mainly due to their growth, which implies a periodic change of prosthesis [11,14]. Constantly purchasing new commercial devices is burdensome for many families [14]. Therefore, 3DP prostheses stand as an alternative, transitory or permanent, for this population [11]. Furthermore, as 3DP prostheses can be personalized with patterns chosen by the young recipients and some of devices are lightweight and easy to actuate, the abandonment rate can be improved in comparison to classic commercial prostheses [1,11,15].

For an individual to wear a 3DP upper limb prosthesis, several essential steps are to be executed carefully. Burn et al. and Tanaka et al. reviewed each of them and reported practical recommendations for choosing the appropriate prosthesis, customizing and printing a device, assembling the printed parts and finally, fitting the prosthesis onto a recipient [3,13]. At each stage, technical obstacles can arise, but the experienced open-source prosthetic community provides valuable support to any individuals around the globe [11,13]. Figure 1 illustrates the successive steps, and their main features, leading to the production of 3D-printed upper limb prosthesis.

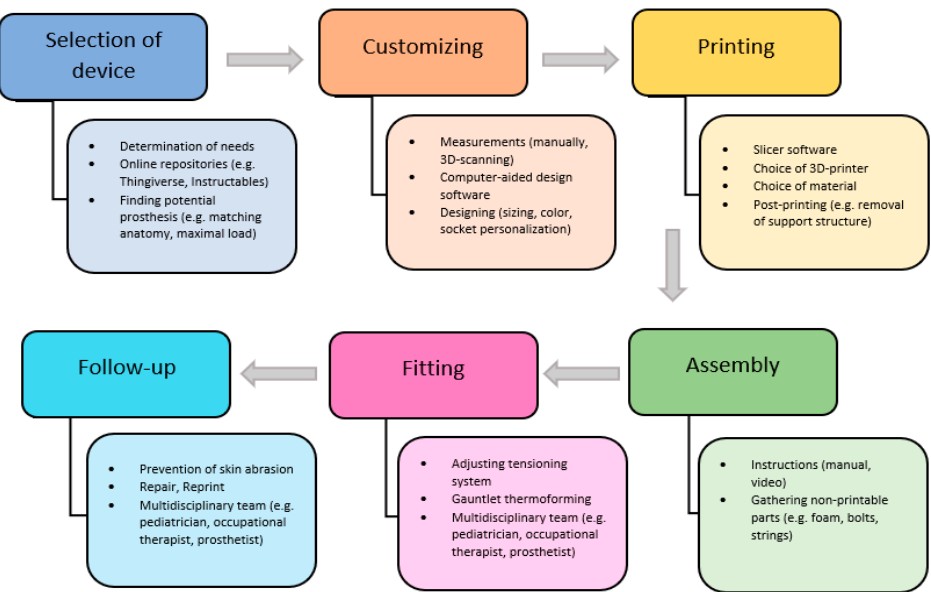

**Figure 1.** Steps involved in the 3D printing of open-source 3DP prostheses.

3D printing technology present some advantages for producing prostheses in comparison to other conventional manufacturing processes (e.g., injection modelling) [16]. First, this additive manufacturing technique allows the fabrication of complex geometric shape and the incorporation of different materials without requiring supplementary tools, production machines or fixturing [11,16,17]. These features enable a designer to customize a 3DP prosthesis for a stronger acceptance and to better fit the 3D-printed device to the recipient's morphology [11]. A recipient-centered approach is therefore essential to successfully answer a user's demands [1]. This technology also permits designers to produce end-products out of one single part with no assembly stage required [17]. Moreover, the production time, depending on the 3D printing technology and 3D printers, can be reasonably limited which allows rapid prosthetic designs adjustments and reprints [16,17]. Finally, the production cost of 3D-printed parts is often reported as an advantage of this technology [11]. Nevertheless, though often presented as a cheap technology, 3D printing objects can be expensive in reality [11,16]. Indeed, 3D printers are often the expensive element of the production line, with prices in the hundreds, and even thousands, of dollars, but that financial aspect is often omitted in reports [11]. Moreover, some 3D printing techniques (e.g., stereolithography) are costlier than others (e.g., fused deposition modelling) [16,17]. Furthermore, some traditional manufacturing processes (e.g., injection modelling) can produce objects at lower prices than 3D printing, and they appear to be more cost and time effective for mass production [11,16]. However, as less expertise and machinery are required for 3D printing parts, it could preferably be the first choice for limited quantities of 3DP parts and for on-demand requests [16].

Nevertheless, most reported limitations to the implementation of this technology in the prosthetic field and to the undertaking of solid comparison studies were: the absence of medical supervision of most 3DP upper limb prostheses available online, the few scientific literature on this subject and the scarce reported data concerning devices' mechanical and kinematic properties which varies widely depending on other parameters (e.g., material, infill, 3D printer) [10,11]. Additionally, the lack of objective data regarding the durability and the possibility of repairing printed parts is also problematic [11,17]. Indeed, through 3D printing, the properties of end-products can differ from those of raw material and environmental conditions (e.g., humidity, UV light, moisture, dust) can influence material degradation [11,17]. Therefore, the longevity of parts is impacted by those factors, yet no data objectify the duration during which a prosthesis can be used before requiring maintenance [11]. Concerning the repair of 3DP protheses, though seemingly simple

to approach, some aspects are important to consider [11]. Most 3DP protheses are not designed and printed by the recipient, who therefore rely on a maker for any reparations or adjustments [11]. The accessibility and availability of a maker to help can potentially lead to unfortunate situations where the prosthetic device cannot be repaired in a reasonably short period of time [11]. A recipient ends up not wearing the prosthesis. Moreover, as multiple additive manufacturing methods can be employed to produce 3DP prostheses, which differ in accessibility and affordability, the rapidity and costs to print spare parts of a prosthesis will be impacted [11]. Additionally, very few data exist concerning the adequacy between recipients' needs and 3D-printed prostheses currently available [11,13]. Furthermore, to this day, there is no study proving their clinical effectiveness, no standardized protocol of production and no validated or available clinical guidelines for their use [10].

However, some groups attempted to find and gather 3D-printed upper limb prostheses. Diment et al. performed a systematic review on this subject, and because of the limited works and their poor methodological quality, they could not conclude on any statistically significant efficacy of upper limb 3D-printed prostheses [10]. The authors also highlighted the lack of data concerning characteristics of devices [10]. Ten Kate et al. reported in 2017 a list of 58 prosthetic devices from both scientific literature and the Internet and extracted data related to their mechanical, kinematic and printing properties [11]. Vujaklija et al. also reported multiple 3DP prostheses of different upper limb segments mentioning their added values [14]. While those two studies valuably contributed to portray an overview of the current situation, they lacked clear methodology in finding and selecting 3D-printed upper limb prostheses.

We hypothesized that the amount of prostheses considered reliable would be reduced if few criteria were determined, possibly due to the absence of clear reporting guidelines, the transiency of projects involving 3D printing technology and the existence of multiple hosting platforms which causes the presence of numerous duplicates.

This study aims to portray the current situation of open-source 3D-printed upper limb prostheses by attempting to reach all devices currently accessible according to an established methodology. We determined criteria to virtually assess the accessibility of those prostheses as no specific protocols exist. We focused only on open-source devices as they are available to all potential recipients and scientific teams worldwide. The data collected would be an aid to further works to improve access, functionality and collaboration in the 3D-printed prosthetic field.

## 2. Materials and Methods

We attempted to gather almost all available open-source, or freely accessible, ready-to-use 3DP upper limb prosthetic devices from both scientific and online sources through different levels of investigations.

Throughout those investigations, we confronted our findings with the following selection criteria:

Inclusion criteria:

- The design and its printing files must be accessible through an open-source license or freely.
- The availability of a clear and reliable support for printing and assembling (manual or video of instructions) must be provided. As both English or French are mastered by the authors, we included devices with information delivered in those two languages.
- The design and its related printing files should have been made accessible after 2018. If the design is anterior to this date but a proof of continuing support (i.e., answers to questions) until at least 2018 is verified, then the device would be included.

Exclusion criteria:

- A design and its related printing files are not accessible through an open-source license or freely.
- A design is available but there is a lack of clear and reliable instructions (i.e., manual, video, etc.) to print or to assemble a device.

- A design is indicated as accessible, but no related printing file is available or the printing files accessible are corrupted.
- A design is indicated by its creator as unsuitable for a daily use or for a purpose different than research.
- Any designs comprising an uncomplete, abandoned or aborted projects. We define an "uncomplete" project as a project, proposed as an accomplished concept or as a "ready-to-use" device, in which there are missing elements (i.e., partial or absence of instructions, code for electronic components, projects steps, etc.). Abandoned or aborted projects are defined as projects not considered accomplished either by its author or by the absence of sign of pursued development on a specific device.

Concerning the sequential investigation stages, we first assessed the review produced by Ten Kate et al. [11], we then analyzed all the scientific papers citing that review, and we finished by searching online databases and repositories hosting 3D printing designs and their related printing files. These steps were performed by authors, K.W. and R.O. Any uncertainties or disagreements were resolved after discussion and unanimous consent between those two authors.

### 2.1. Ten Kate et al. Review

First, we decided to determine whether the data reported in the first robust review of 3D-printed upper limb prostheses, from Ten Kate et al. in 2017, were still accurate [11]. To accomplish this, we reviewed all the 3DP upper limb prostheses reported in that review to assess their actual existence and collect basic information. For devices appearing as still available and verified as accessible, we subjected them to our selection criteria.

We so assessed each of the 58 prostheses discussed in the review from Ten Kate et al. [11]. For clarity, we extracted the following data for each device: source (i.e., scientific literature or online 3DP database), link (literature reference or the URL), author(s), date of upload or publishing, picture, instructions media (manual, video), and presence of support (help provided to the users (e.g., answers in forums)), and accessibility (i.e., free access to and download of 3D printing files). If a device was not available, we recorded the cause.

We then classified both included and excluded devices in categories according to the final decisions.

### 2.2. Literature Review

We secondly performed an oriented literature review in order to collect the open-source 3D-printed upper limb prostheses reported in the scientific literature. Indeed, we reviewed all published materials citing the review from Ten Kate et al. [11].

In addition to the previously mentioned selection criteria, we set the condition of accepting only peer-reviewed articles and excluded other types of publications (e.g., books, etc.). No handsearch (i.e., conference reports) was undertaken and other sources (i.e., reference lists) searched. Those articles were to be written either in English or French as both languages are mastered by the authors. All applications discussed were accepted as long as those criteria were respected. Nevertheless, contrary to the inclusion criteria, here, there was no date restriction applied. The last search was performed on the 11th of January 2021.

We screened all records based on their titles and abstracts. Secondly, we reviewed the full-text of each included publication and subjected them to the selection criteria previously described. For each article, the investigator was aware of its publication details (e.g., authors, date, journal, etc.).

We also extracted the following data for their 3D-printed prostheses: name, creator(s), original hosting database, year of upload, provision of instructions, and mechanical and kinematic specifications.

We reported the studies included with their main characteristics and findings. No statistical analysis was performed.

*2.3. Online Databases Search*

We concluded our attempt to gather all 3D-printed upper limb prosthetic devices by searching online databases. We chose those online repositories based on their specialization in 3D printing as they host 3D-printable designs and their associated printing files. We employed terms related to 3D printing and upper limb segments. Here, are the online repositories searched: Thingiverse, Cults, CGTrader, MyMiniFactory, Pinshape, TurboSquid, PrusaPrinters, 3DExport, YouMagine, NIH 3D Print Exchange, Free3D, Redpah, XYZprinting 3D Gallery, Fab365, Instructables, Zortrax library and Libre 3D.

We subjected each online database and their devices to the selection criteria and extracted the following data for included 3D-printed prostheses: name, creator, original hosting database, year of upload, provision of instructions, signs of continuing support, signs of further development, and mechanical and kinematic specifications.

e-NABLE Platform

We thoroughly searched the online e-NABLE platform as it is a living global community that aims to give free access to 3D-printed upper limb prosthetics and orthotics to individuals, young and old [18]. Their specialized repository gathers multiple 3D-printable upper limb prosthetics which are hosted by other external online databases. Thus, it was predictable that some duplicates would be found between the devices they supported and those present on original online repositories.

We only reviewed the devices categorized under the sections "Arms Designs" and "Hands Designs" on the webpage 'e-NABLE Devices Catalog' [19]. All the arm and hands devices assessed are gathered in Table 1.

**Table 1.** Upper limb prosthetic designs from e-NABLE Devices Catalog.

| **Arm Designs** |
| --- |
| "El Medallo" Bionic Arm |
| Adjustowrap Gripper Arm |
| Flexy Arm |
| Kwawu Arm |
| Phoenix Reborn Arm |
| Po Arm |
| Self-suspending below-elbow sockets methodology |
| Unlimbited Arm v2.1 |
| Versatile Elbow Operated Gripper—VEOG |
| **Hand Designs** |
| Cyborg Beast |
| e-NABLE Phoenix Hand v3 |
| Flexy-Hand 2 |
| K1 Hand |
| Kinetic Hand |
| MotoGripper Terminal Device |
| Ody Hand |
| Osprey Hand |
| Phoenix v2 Hand |
| Raptor Reloaded |
| Talon Hand |
| The Paraglider |
| Unlimbited Phoenix Hand |

It is relevant to indicate that the e-NABLE community rates the devices present in its catalog based on five categories: maturity, cost of materials, popularity, difficulty and grip strength. For full explanation, their Device Ratings Guide is available online [20]. We specifically decided to add the "Maturity" criteria to screen devices from their platform. An upper limb prosthetic device with a maturity considered "High" refers to a 3D printable

prosthesis associated with a solid documentation and testing background. Therefore, we included any devices which maturity was considered as "High" and beneficiating from an ongoing provision of technical support by the e-NABLE community, even though the prosthetic designs might have been uploaded before 2018 on their original database and possibly with no recent updates on their printing files.

## 3. Results

We organized the results according to the successive levels of investigation. The flowchart in Figure 2 illustrates a summarized view of the sequential search process and the main findings.

**n = 2**
- Assessment of devices from ten Kate et al. review
- 58 prostheses screened

**n = 6**
- Literature review
- 166 publications screened

**n = 25**
- Online repositories hosting 3D-printable designs
- 17 online databases searched

**n = 12**
- e-NABLE community platform search
- 22 devices screened

- 6 articles discussing applications of open-source 3DP upper limb prostheses included.
- 39 open-source 3DP upper limb prostheses included.

**Figure 2.** Study investigations flow.

### 3.1. Review of Ten Kate et al.

We confronted all 58 devices reported in the review performed by Ten Kate et al. with the criteria mentioned previously [11]. From those 58 prostheses, 7 were from scientific articles and 51 from online websites and repositories. Out of that analysis, only 3 open-source 3D-printed prostheses, among the 51 ones reported from online sources, met our selection criteria and were included in our study, namely, K-1 hand, The Cyborg Beast hand and Flexy hand 2 [21–23]. They are all hand prostheses and hosted on both an online repository (i.e., Thingiverse or NIH 3D Print Exchange) and the e-NABLE organization platform.

Concerning the other devices hosted on online databases, which were excluded, we classified them according to the reason of their exclusion. See Table 2 for the full classification.

**Table 2.** Prosthetic devices reported in Ten Kate et al. review (from online sources).

| **Excluded Devices** | | | | | | | | | | |
|---|---|---|---|---|---|---|---|---|---|---|
| *Hosting website not reachable (not functioning URL)/No information about the actual device on the hosting website.* | | | | | | | | | | |
| Zero point Frontiers | One-hinged Cyborg Beast | Cyborg Beast with Increased Wrist Movement | JD-1 | NuHand | IVINA 2.0 | Handiii Coyote | Handiii | One-hinged Cyborg Beast | Adjustable Thumb | The Cyborg arm |
| *Not Open-source/Printing files not accessible/In press only.* | | | | | | | | | | |
| Youbionic | Victory Hand | Tenim Hand | Protesis Cosmetica | Not Impossible | Manu Print (Re Hand) | Hero arm (Bionic arm) | 3D-printed prosthesis Ecuador | Bionico | | |
| *Absence of reliable instructions (manual, video ... ) for specific customization, 3D printing or assembling* [†]*.* | | | | | | | | | | |
| Robot Hand | Robohand | InMoov 2 hand | Hollies Hand Version | Flexy arm | | | | | | |
| *Posted before 2018 and lack or no more support/No sign of further development* [‡]*.* | | | | | | | | | | |
| The Lucky Paw Prosthetic Hand | Talon Flextensor 1.0 | Hackberry | Flexy hand | e-NABLE RIT Arm | Muscle Robot Hand | GalileoHand | Limbitless Arm | Dextrus EMG | DIY Prosthetic Hand and forearm | Falcon Hand V2 |
| *Lack of data or information for producing functioning devices (e.g., Printing files, Arduino code, etc.).* | | | | | | | | | | |
| Tact: Low-cost, advanced prosthetic hand | Scand hand | Roboarm | Mind controled Robot Hand | | | | | | | |
| *Aborted projects.* | | | | | | | | | | |
| Biohand | | | | | | | | | | |
| *Devices referred by creators as suited research purpose only or not suited for patients or extended use.* | | | | | | | | | | |
| Snap-Together Robohand | e-NABLE Raptor Reloaded | e-NABLE Raptor Hand | Falcon Hand V1 | | | | | | | |
| **Included devices** | | | | | | | | | | |
| K-1 | The Cyborg *Beast | Flexy hand 2 | | | | | | | | |

Caption: [†]: 'Customization' is defined as the personalization of the digital model to fit the recipient's residual limb. [‡]: 'no more support' is defined as the absence of response to support requests concerning the customization, 3D printing or assembly of devices on online platforms (i.e., unanswered questions). 'no sign of further development' is defined as the absence of any proof of any improvements made on a device.

In addition to those 51 devices from online databases, Ten Kate et al. pointed out seven scientific works discussing the development of 3D-printed upper limb prostheses. None of these seven studies was included: one was a not-peer-reviewed doctoral thesis which consisted of the design of a prosthetic finger by Groenewegen et al. and all the six remaining studies did not involve any open-source, or freely accessible, 3D-printed prostheses [24–30].

### 3.2. Review of Literature

From our search, 166 publications were found. After screening those findings based on titles and abstracts, only 56 papers were included for a full-text review. From that step, only six articles were included in our study [31–36]. Among those six studies, two assessed the functionality of specific prostheses [31,32], two were case studies [33,34] and the last two ones were an original research article and one cohort study [35,36], respectively.

As the aims and settings of those studies differed, we gathered their main characteristics and results in Table 3.

**Table 3.** Studies implementing open-source 3D-printed prostheses.

| Author | Year | Number of Prostheses Studied | Name of Device | Level of Prosthetic | New vs. Existing Open-Source Prosthesis | Motion | Study Aim | Study Setting | Number Participants | Main Results | Availability of Designs |
|---|---|---|---|---|---|---|---|---|---|---|---|
| Alturkistani et al. | 2020 | 1 | Raptor Reloaded Hand | Hand prosthetic | Existing prosthesis | Passive | Developing affordable partial hand prosthesis with flexible material | Design process with patient's active participation Qualitative assessment (questionnaire) Quantitative assessment (grasping test, lift test) | 1 (trans-metacarpal amputation, missing three fingers) | Grips by using contralateral hand Low grip strength (700 g) but function considered as sufficient by participant (stable grasp) Bimanual activities achievable. | Online repository † |
| Anderson et al. | 2021 | 1 | Talon hand | Hand prosthetic | Existing prosthesis | Active (wrist) | Developing a 3DP hand prosthesis allowing a child participation in gymnastic class | Impact assessment through testing specific gymnastic skills and questionnaire | 1 (left hand with congenital deficiency) | Improvement in performing specific gymnastic classes; Increased satisfaction, confidence, participation in gymnastic classes. | Online repository † |
| Neethan et al. | 2019 | 4 | Flexy hand, Shira, Limbforge, "bionic hand" | Hand prosthetic | Existing prostheses | Active (wrist) | Comparison of strength, comfort and production cost | Analysis for Flexy and Shira hands: Functionality testing (grip strength, grasping); comfort analysis; production cost estimation | 0 | Flexy hand most suit-able/appropriate device as lesser cost, reduced effort requirements for users. | Online repository |
| Omar et al. | 2019 | 1 | HACKberry | Hand prosthetic | Existing prosthesis | Active (myoelectric) | Developing 3DP prosthetic bionic hand with appropriate sensory and control tuning to perform basic activities of daily living (ADL) | Calibration, Grasping and ADL test. | 5 (not amputees) | Limited Hand functionality (limited grasps options, slippery surface). | Online repository |
| Tong et al. | 2019 | 1 | Raptor Hand | Hand prosthetic | Existing prosthesis with additional integrated 3DP electrodes with pressure sensors | Active (wrist) | Approach to create a personalized 3DP hand prosthesis with integrated 3DP electrode for measuring pressure distribution | 3D scanning, reverse engineering, design process, 3DP of integrated electrodes, analysis of pressure distribution on upper limb | 1 (amniotic band syndrome, right hand) | Association of 3D scanning and 3D printing enables creation of form-fitting 3DP personalized low-cost hand prostheses with integrated electronic components. | Online repository † |
| Zuniga et al. | 2017 | 1 | Cyborg Beast 2 | Hand prosthetic | Existing prosthesis | Active (wrist) | Analysis of functional and strength changes after usage of a 3DP transitional prosthesis in children with upper limb difference | Function testing (Box and Block Test) Strength measurements (strength testing with dynamometer) | 11 (Congenital defects, amputation) | Improvement of manual gross dexterity (function); no significant impact on strength of residual wrist. 3DP prosthesis can be used as a transitional device to improve function. | Online repository † |

†: The final personalized design version is not available on online repositories. "Availability of designs" section: The indication 'Online repository' means that the design of the device is hosted on an online database. "Motion" section: It refers to the active or passive capacity of a prosthetic device and if active which residual articulation triggers the motion of the prosthesis.

### 3.2.1. Population

Most studies (4 out of 6) included patients, both pediatric and adult, affected by either congenital or traumatic defects [33–36]. It is noteworthy to indicate that Omar et al. included participants for manipulating and testing the prosthetic devices, who were not suffering from any affection of their respective upper limbs [31]. Additionally, Neethan et al. conducted their comparison study on four upper limb prostheses without involving any participants [32].

### 3.2.2. Design

All devices assessed in those six studies had their original printing files accessible on online repositories [31–36]. Nevertheless, in four studies, an additional digital personalization step was performed but their final customized design was not indicated as obtainable by their respective authors [33–36]. Moreover, it is relevant to note that those six teams employed already existing 3D-printed prostheses and none of them discussed the development of a new freely accessible 3D-printed upper limb prosthesis [31–33]. Nevertheless,

some groups did explore new approaches in order to customize their devices on behalf of the recipient [34,35]. For example, Tong et al. added 3D-printed electrodes containing pressure sensors on their customized 3DP upper limb prosthesis in order to study forces distribution on the affected limb [35]. Alturkistani et al. completely redesigned an original prosthesis in order to fit their participant's amputated hand, which consisted of the 4th and 5th fingers and a partial palm [34].

### 3.2.3. Mechanical Specifications

*Level of prostheses and actuation*. Out of those six studies, four of them employed body-powered prostheses (i.e., with wrist or elbow motion) [32,33,35,36], one single group discussed the efficiency of a myoelectric, externally powered device [31], and one team developed an adjustable passive prosthesis [34].

All body-powered and the only passive devices were hand prostheses [32–36], and the only externally powered prosthesis was forearm-leveled [31]. For the latter, Omar et al. employed an infra-red sensor to detect muscle contraction [31].

*Weight*. Two teams provided indications relative to the weight of their prosthesis [32,34]. Alturkistani et al. indicated that their device weighed less than 100 g and Neethan et al. only specified the weight of plastic used for each prosthesis compared in their study [32,34].

*Maximal Load*. Three groups performed measures to specify the maximal load of their prostheses [31,32,34]: Alturkistani et al. reported the limit of 700 g of load for their prosthetic device [34]; Omar et al. indicated that the Hackberry hand could not lift a load exceeding 2000 g [31]; Neethan et al., who compared four devices and specifically assessed two of these, reported for those two specific prostheses a maximal charge of 3059 g and 2039 g, using a rectangular prism bar and an ovoid bar, respectively [32]. It is noteworthy to indicate that the measures recorded by Neethan et al. were performed by individuals in possession of their both healthy upper limbs [32].

*Actuators*. All 3DP prostheses body-powered, actuated by either a functional wrist or elbow, were therefore constituted of one single actuator [32,33,35,36]. Only the Hackberry hand assembled by Omar et al. possessed three actuators [31]: three servo motors were embedded to actuate fingers movements [31]. No additional technical information on those motors was provided by the authors [31]. The adjustable passive partial hand prosthesis developed by Alturkistani required the contralateral hand to be adjusted [34].

### 3.2.4. Kinematic Specifications

*Range of motion*. Only Zuniga et al. and Neethan et al. reported range of motion values [31,36]. They measured the flexion angles to reach the full closure of their prosthetic devices. The former indicated a 20–30° flexion angle for their prosthesis and the latter measured angles of 46° and 35° for the studied Flexy and Shira hands, respectively.

*Grasping*. Each grasp type is classified by its need for power or precision to be adequately performed. Moreover, the fingers position, thumb adduction or abduction and the objects shape also contribute to this naming and classification. Recent GRASP Taxonomy classifies specific grasps (e.g., lateral, tip, cylindrical, hook, spherical) under those two main groups, power and precision, and a third one, intermediate grasp [37].

Three studies out of six addressed the different grasps each studied prosthesis can perform [31,32,34]. All those four hands could perform power, lateral and precision grips. In addition, tip, cylindrical, palmar and hook grasps were also achieved by some devices [31,32,34]. Anderson et al., who 3D-printed a prosthesis to support a young gymnast [33], Zuniga et al., who studied 3D-printed devices and transitional prostheses [36], and Tong et al., who produced a low-cost sensor-integrated 3D-printed personalized hand prosthesis for children with amniotic band syndrome, did not disclose that information [35]. Omar et al. indicated that their Hackberry hand, though it was capable of performing four main grasps (power, precision, lateral, tip), it could not ensure a sure grip as the palmar surface was reported to be slippery and rigid [31].

No study discussed the degree of freedom (DoF) of their 3D-printed prosthetic devices [31–36].

No study provided data concerning the number of joints of the 3DP prostheses [31–36].

### 3.2.5. 3D Printing Processes and Materials

All six studies used a fused deposition modelling (FDM) 3D printer to print their 3D-printed upper limb prostheses [31–36].

Most teams (4/6) using FDM 3D printers opted for Polylactic Acid (PLA) only [31,33,35,36], a hard-plastic 3D printing material, to print their prosthetic device. It was inconsistently associated with acrylonitrile butadiene styrene (ABS) (1/6) [36], another thermoplastic material, or with flexible material (1/6) such as thermoplastic copolyester (TPC) [32]. One team employed some flexible material only, thermoplastic polyurethane, to 3D-print their final prosthetic device [34].

### 3.2.6. Production Cost

Three teams disclosed the production cost of their devices [31,33,34]. Omar et al. indicated an estimated production fee of USD 449 and these could reach USD 862 if a new 3D printer was to be acquired [31]. Anderson et al. reported that their prosthesis cost less than USD 40 and for theirs, Alturkistani et al. indicated manufacturing costs ranging from USD 20 to USD 25 for each device printed along their iteration process [33,34].

### 3.2.7. Functionality Assessment

Two studies attempted to evaluate the functionality of prostheses [31,36]. On one hand, Zuniga et al. studied the impact of their Cyborg Beast 2 prosthesis as a transitional prosthesis for children [36]. They observed, through the Box and Block test, that the gross dexterity of the affected limb was improved after 24 weeks of use, but that the strength of the residual wrist was not significantly impacted by the wear of the device. On the other hand, Omar et al. approached the efficacy of the studied Hackberry prosthesis by analyzing its ability to perform specific activities of daily living (ADL) [31]. Activities of daily living are the essential, both basic and complex, tasks of self-care in daily life to be performed by an individual in order to be considered independent, and specific grasping is required for each ADL [11,38]. Table 4 includes the ADLs assessed by Omar et al. [31]. They demonstrated that their device could only realize four of those seven ADL: tying shoelace, lifting a water bottle (600 mL) to mouth height, picking up a shirt and turning a book page. The participants were not able to open a door nor picking up a jacket. Finally, picking up a pen and writing was possible but associated with many difficulties [31].

**Table 4.** Activities of daily living assessed by Omar et al. [31].

| |
|---|
| Lifting a 600 mL water bottle up to the mouth. |
| Opening a door. |
| Picking up a pen and writing. |
| Picking up and holding a jacket from a point A to a point B. |
| Picking up and holding a shirt from a point A to a point B. |
| Turning a page of a book. |
| Tying shoelaces. |

Anderson et al. and Alturkistani et al. submitted a satisfaction survey to their recipient participants [33,34]. The former reported that the young gymnast recipient's responses indicated an improvement in activities participation, self-confidence and satisfaction in gymnastics practice with the prosthesis [33]. Concerning the passive prosthetic device studied by Alturkistani et al., the participant reported, as main advantages, the compactness and the low weight of the prosthesis as well as the easiness to put it on quickly [34]. Moreover, the device also provided enough stability allowing some bimanual tasks [34].

### 3.3. Review of Online Databases

Consecutively to investigating the review from Ten Kate et al. and related articles, we explored online specialized repositories hosting 3D printable objects.

### 3.3.1. General Repositories

From the seventeen online repositories searched, we found open-source upper limb prostheses' designs, meeting our selection criteria, on only two of them, namely, Thingiverse and Instructables [39,40]. These are two online platforms, freely accessible, allow designers to share open-source 3DP projects and their related printing files.

### 3.3.2. Design

We found 25 upper limb prosthetic devices meeting our selection criteria, 24 on Thingiverse [23,41–63], and 1 on Instructables [64]. See Table 5 for the complete list of included devices. We also shared examples of open-source 3D-printed upper limb prostheses in Figures 3 and 4.

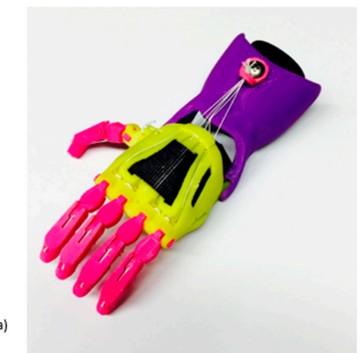

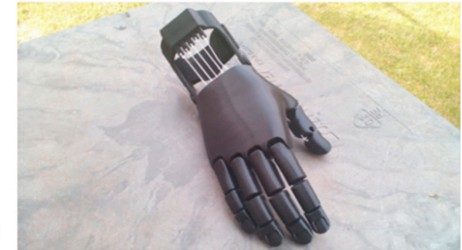

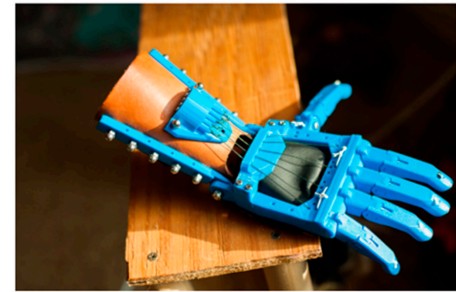

**Figure 3.** Examples of open-source 3D-printed hand prostheses. (**a**) The Cyborg Beast. Creator: Jorge Zuniga. Original source: https://www.thingiverse.com/thing:261462 (accessed on 3 May 2022). No changes made. Reproduced under creative commons attribution 4.0. No permission required [22]. (**b**) Flexy Hand 2. Creator: Gyrobot. Original source: https://www.thingiverse.com/thing:380665, (accessed on 5 May 2022). No changes made. Reproduced under creative commons attribution 4.0. No permission required. [23]. (**c**) Talon Hand 3.0. Creator: Profbink. Original source: https://www.thingiverse.com/thing:229620, (accessed on 3 May 2022). No changes made. Reproduced under creative commons attribution: no copyright (Public Domain Dedication). No permission required [50].

**Table 5.** 3D-printed upper limb prostheses from online databases.

**Hand Prostheses**

Source: Thingiverse

| Name | Creator | Year | Actuation | Versions | Instructions | Support | Progress | Comments |
|---|---|---|---|---|---|---|---|---|
| Cathy's Lucky Fin V3—Prosthetic Hand—Bowden/Push-Pull Variant | Rhadamanthys76 | 2021 | BP [‡] | 3 | Yes | Yes | / | Support provided up to 2020; instructions more complete in versions 1 et 2. |
| Flexibone Prosthetic Hand 2019 | TeamGrenable | 2019 | EP [‡] | 1 | Yes | N/A [†] | / | / |
| Flexy-Hand | Gyrobot | 2014 | BP | 2 | Yes | Yes | Updated in 2015 | Support provided up to 2021. |
| Flexy-Hand 2 | Gyrobot | 2014 | BP | 2 | yes | Yes | / | Support provided up to 2018. |
| Flexy-Hand 2—Filaflex Remix | Gyrobot | 2015 | BP | 1 | Yes | Yes | / | Support provided up to 2018. |
| Gold Dexterity Hand | Nickhs | 2018 | BP | 1 | Yes | Yes | Updated in late 2018 | / |
| Modular Flexy Hand 2 (Interchangeable fingers) | HHP_UNCC | 2019 | BP | 1 | Yes | Yes | Updated in late 2019 | / |
| Ody Hand 2.1 | Profbink | 2014 | BP | 2 | Yes | Yes | Updated in 2018 | Support provided until this day. |
| Phoenix Talons | HHP_UNCC | 2019 | BP | 1 | Yes | N/A | Updated in late 2019 | / |
| Robotic prosthesis | Bfessler | 2019 | EP | 1 | Yes | N/A | / | / |
| Talon Hand 3.0 | Profbink | 2014 | BP | 3 | Yes | Yes | Updated in 2017 | Support provided until this day. |
| The Osprey Hand by Alderhand and e-Nable | Profbink | 2015 | BP | 1 | Yes | Yes | Updated in 2018 | Support provided until this day. |

Source: Instructables

| | | | | | | | | |
|---|---|---|---|---|---|---|---|---|
| Servo-Controlled Prosthetic Hand | Duncanlaird | 2018 | BP | 1 | Yes | N/A | / | / |

**Forearm prostheses**

Source: Thingiverse

| Name | Creator | Year | Actuation | Versions | Instructions | Support | Progress | Comments |
|---|---|---|---|---|---|---|---|---|
| Arm v2 | Masnart39 | 2015 | BP | 2 | Yes | Yes | Updated in 2016 | Support provided until this day. Printing files available in different formats. |
| Bionic Flexy Arm II | Materializacion3DColombia | 2016 | BP | 1 | Yes | Yes | Updated in 2019 | Instructions in video. |
| Cosmetic lower arm prosthetic | Hatsyflatsy | 2019 | /¥ | 1 | Yes | N/A | Updated in 2020 | Limited instructions. |
| E-Talon | 1d1 | 2019 | EP | 1 | Yes | N/A | / | / |
| Kwawu Arm 2.0—Prosthetic | JacquinBuchan | 2018 | BP | 3 | Yes | Yes | Updated in 2019 | / |
| Kwawu + Rojava Remix Arm Prosthetic | Mimi_3d | 2021 | BP | 2 | Yes | N/A | / | / |
| My Customized The UnLimbited Arm v2.1—Alfie Edition | Edoubleb | 2017 | BP | 1 | Yes | Yes | / | Answers up to 2018, referring to UnLimbited Arm assembly instructions. |
| NIOP Kwawu remix | NateMunro | 2019 | BP | 1 | YES | Yes | / | / |
| Prótesis personalizada Cinderella (cenicienta) | Materializacion3DColombia | 2016 | BP | 1 | Yes | N/A | Updated in 2019 | Instructions in video. |
| Robotic Prosthetic Hand | Grossrc | 2016 | EP | 1 | Yes | Yes | Updated in 2019 | Support provided up to 2019. |
| Unlimbited FP3D | FundacionProtesis3D | 2017 | BP | 1 | yes | N/A | Updated in 2021 | Instructions in video. |

**Arm prostheses**

Source: Thingiverse

| | | | | | | | | |
|---|---|---|---|---|---|---|---|---|
| Robo arm | Cloudyconnex | 2021 | UD [‡] | 1 | Yes | N/A | / | / |

Caption: [†]: N/A refers to the absence of support request on an online database for that specific device when our search was performed. [‡]: BP: Body-powered; EP: Externally powered; UD: Undefined. ¥: This prosthetic device is a cosmetic prosthesis with no actuation. "Comments" section: The indication "Original platform" means that the technical support is predominantly provided on the original database of the device. "Progress" section: Date of any updates made on the device (designs, printing files, instructions, printing recommendations, etc.).

### 3.3.3. Mechanical Specifications

*Level of prostheses and actuation.* Thirteen are hand prosthetic [23,41–51,64], eleven are forearm devices [52–62], and one is an arm device [63].

Except for the Flexibone Prosthetic Hand and the Robotic prosthesis which are externally powered and electrically powered [42,49], all remaining eleven hand prostheses are body-powered with wrist motion [23,41,43–48,50,51].

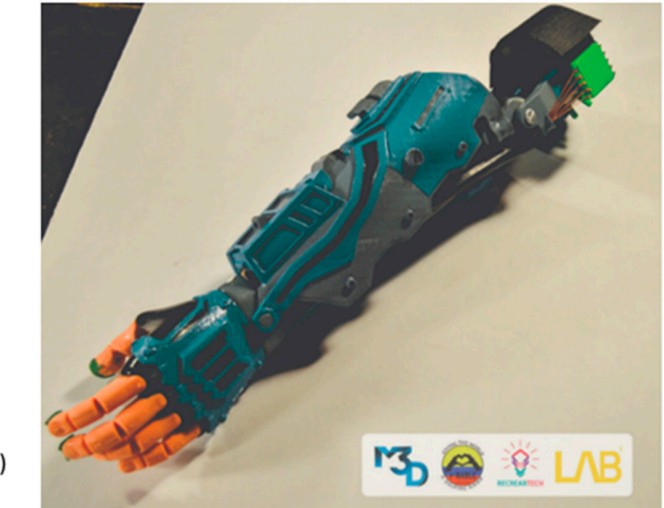

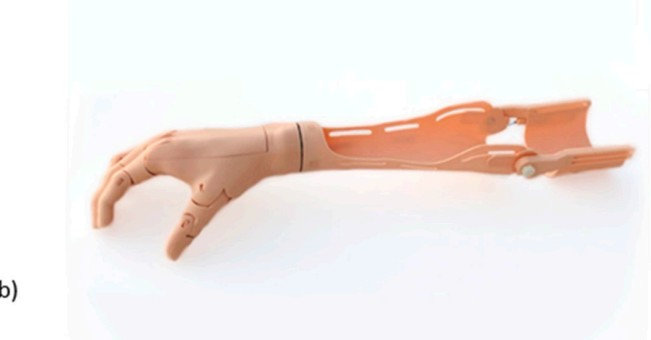

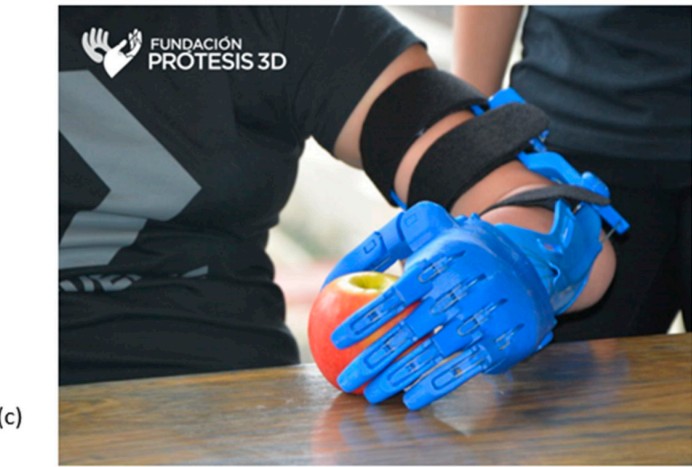

**Figure 4.** Examples of open-source 3D-printed forearm prostheses. (**a**) Bionic Flexy Arm II. Creator: Materializacion3DColombia. Original source: https://www.thingiverse.com/thing:1768698 (accessed on 3 May 2022). No changes made. Reproduced under creative commons attribution 4.0. No permission required [53]. (**b**) Kwawu Arm 2.0—Prosthetic. Creator: Jacquin Buchanan. Original source: https://www.thingiverse.com/thing:2841296, (accessed on 3 May 2022). No changes made. Reproduced under creative commons attribution 4.0. No permission required [56]. (**c**) Unlimbited FP3D. Creator: FundacionProtesis3D Original source: https://www.thingiverse.com/thing:2564128, (accessed on 5 May 2022). No changes made. Reproduced under creative commons attribution 4.0. No permission required [62].

In addition, one passive static prosthesis designed for cosmetic purposes the Cosmetic lower arm prosthetic [54], all the forearm prostheses are actuated [52,53,55–62,64]. There are eight body-powered, elbow-actuated, forearm prostheses [52,53,56–60], and two externally powered, the Robotic Prosthetic Hand and the E-Talon prostheses, both are electrically powered [55,61].

The type, active or passive, and the actuation are not specified concerning the only arm prosthesis included, the Robo Arm [63]. See Table 5 for detailed information.

*Actuators.* All 3DP prostheses body-powered, actuated by either a functional wrist or elbow, were therefore constituted of one single actuator [23,41,43–48,50–53,56–60,62]. The four externally powered prostheses comprised additional actuators [42,55,59,61,63]: the Flexibone Prosthetic Hand and the Robotic Prosthetic Hand embedded two servo motors to actuate fingers movements [42,61]; the Robotic prosthesis device incorporates five servo motors [49]; the type of actuators used in the E-Talon prosthesis is not clearly reported [55]. Only the creators of the Flexibone Prosthetic Hand provided the technical specifications of the servo motors [42].

*Weight.* Only one creator reported indications relative to the weight of a prosthetic devices on the hosting online repository: the NIOP Kwawu remix which weights 737 g [59].

*Maximal Load.* Information concerning the strength of a device was only mentioned for one single prosthesis, the Talon Hand. Its maximal load was reported to approximate 13 lb (±6 kg) [50]. To illustrate this, the creator shared a video showing a recipient lifting a 13 lb (±6 kg) dumbbell with a left 3D-printed Talon Hand [50].

### 3.3.4. Kinematic Specifications

*Range of motion.* No creator shared data concerning range of motion nor degree of freedom.

*Grasping.* Only the Kwawu Arm 2.0—Prosthetic device is provided with information relative to its possible grasps [56]. In an explanatory video, its maker, Jacquin Bachanan, mentioned some power, precision, tripod and pinch grips [56].

No creator provided data concerning the number of joints of the 3DP prostheses [23,41–64].

### 3.3.5. Production Cost

No information related to the production cost was provided for any devices [23,41–64]. See Table 5 for the full characterization of those devices.

### *3.4. e-NABLE Platform*

### 3.4.1. Design

We only reviewed the devices categorized under the sections "Arms Designs" and "Hands Designs" on the webpage 'e-NABLE Devices Catalog' [19]. We analyzed 9 arm and 13 hand prosthetics, respectively. As previously mentioned, the arm and hands devices assessed are gathered in Table 1.

Concerning the arms designs, from those nine prostheses, only three were included: Unlimbited Arm v2.1, "El Medallo" Bionic Arm, Kwawu arm [56,65–69]. We gathered their main characteristics in Table 6.

The Flexy and Po arms were excluded as they were not associated with clear and reliable instructions and support [70,71]. The Adjustowrap Gripper Arm was excluded as its related printing files were not accessible [72]. The Self-suspending below-elbow sockets methodology and Versatile Elbow Operated Gripper—VEOG—designs were also excluded as they were socket-related and not anthropomorphic, respectively [73,74].

Secondly, concerning the hand prosthetic devices group, only nine of them were included [75–87]. Refer to Table 6 for the complete list of included hand prostheses and their characteristics.

**Table 6.** 3D-printed upper limb prostheses from the e-NABLE platform.

| | | | | **3D-Printed Forearm Prostheses** | | | | |
|---|---|---|---|---|---|---|---|---|
| **Name** | **Creator** | **Original Hosting Platform** | **Year** | **Maturity** | **Instructions** | **Support** | **Progress** | **Comments** |
| "El Medallo" Bionic Arm | eNABLE Medellin (Mark Walbran et al.) | Github | 2018 | High | Yes | Yes | / | Original platform |
| Kwawu arm | Jacquin Buchanan | Github | 2018 | Medium | Yes | Yes | Update in 2019 | Original platform |
| Unlimbited Arm v2.1 | Team Un-Limbited | Thingiverse | 2017 | High | Yes | Yes | / | Original platform |
| | | | | **3D-printed Hand prostheses** | | | | |
| e-NABLE Phoenix Hand v3 | Jason Bryant et coll. | Thingiverse | 2019 | High | Yes | Yes | / | Original platform |
| Flexy hand 2 | Gyrobot team | Thingiverse | 2014 | High | Yes | Yes | / | e-NABLE |
| Kinetic Hand | Mat Bowtell | Thingiverse | 2020 | High | Yes | Yes | / | Original platform |
| Ody Hand | Peter Binkley | Thingiverse | 2014 | High | Yes | Yes | Update in 2018 | Original platform |
| Osprey Hand | Peter Binkley | Thingiverse | 2015 | High | Yes | Yes | Update in 2018 | Original platform |
| Phoenix v2 Hand | Jason Bryant et coll. | Thingiverse | 2016 | High | Yes | Yes | / | e-NABLE |
| Talon Hand 3.0 | Peter Binkley et coll. | Thingiverse | 2014 | High | Yes | Yes | Update in 2017 | Original platform |
| The Cyborg Beast | Zuniga et al. | Thingiverse | 2014 | High | Yes | Yes | / | e-NABLE |
| Unlimbited Phoenix Hand | Team Unlimbited | Thingiverse | 2017 | High | Yes | Yes | / | e-NABLE |

Remarks: "Comments" section: The mention "Original platform" indicates that the technical support is predominantly provided on the original database of the device. The indication "e-NABLE" refers to the absence of follow-up on the original database but with a support provided by the e-NABLE community on the e-NABLE online platform. "Progress": Date of any updates made on the device (designs, printing files, instructions, printing recommendations, etc.).

Out of the nine included hand prosthetic devices, two hand designs, the Ody Hand and the Cyborg Beast, both from 2014, were still associated with the same old printing files but are indicated as "mature" designs by the e-NABLE rating system and so were included [76,77]. Additionally, it appeared that for four of these nine devices, the Cyborg Beast, Flexy hand 2, Unlimbited Phoenix Hand, and Phoenix v2 Hand, there was no more assistance provided to users on their original platforms (i.e., Thingiverse) for many years in some cases but it persisted on e-NABLE platform [77,79,84,86].

The four rejected 3DP hand prostheses were namely the Raptor Reloaded (research-purposes only), The Paraglider (no sufficient instructions), the MotoGripper Terminal Device and Forefinger Gripper Hand (no socket associated) [88–91].

It is noteworthy to highlight that the arm designs are relatively recent in comparison to the hand designs. Indeed, the oldest arm design is dated in 2017 (i.e., the Unlimbited Arm v2.1) versus 2014 for the hand prosthetic ones (e.g., The Cyborg Beast) [22,66].

### 3.4.2. Mechanical Specifications

*Level of prostheses and actuation.* Both the Unlimbited Arm v2.1 and Kwawu arm were forearm prostheses, body-powered and actuated by elbow motion [56,66]. The "El Medallo" Bionic Arm is an externally powered forearm prosthesis, electrically powered, and controlled by EMG-sensors [68]. All included 3DP hand prostheses were body-powered, actuated by a functioning wrist [75–87].

We have gathered in Figure 5 the multiple types of actuation for the different levels of all the prostheses included in our study.

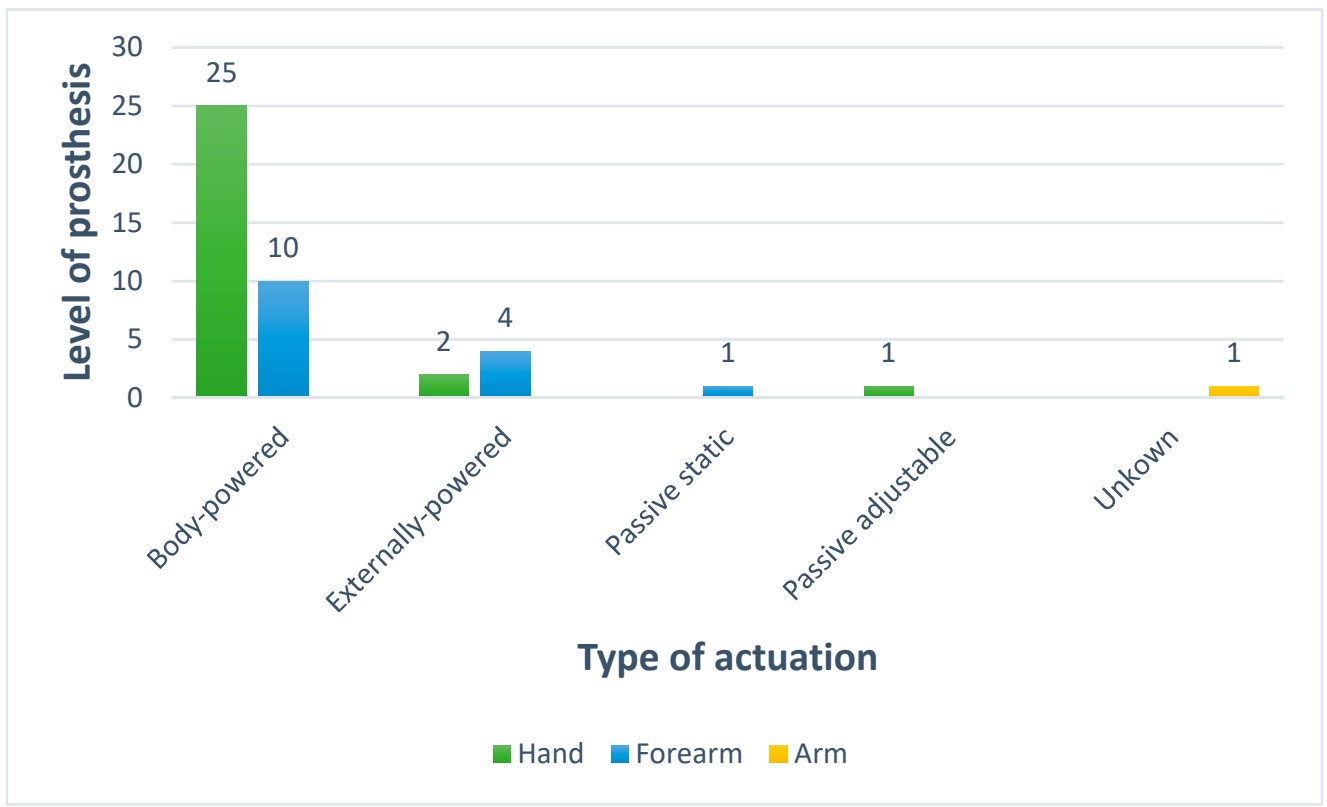

**Figure 5.** Types of actuation and levels of prostheses.

*Actuators.* All 3DP body-powered prostheses, actuated by either a functional wrist or elbow, therefore comprised one single actuator [56,65–69,75–87]. The only externally powered prosthesis, the "El Medallo" Bionic Arm, comprised additional actuators [68]: it embedded two servo motors to actuate fingers movements. The technical specifications of the servo motors were not provided [68].

*Weight.* No information related to the total weight of a device was provided for any prostheses [56,65–69,75–87].

*Maximal Load.* Except for the Talon Hand, for which the load limit was previously indicated, the "El Medallo" Bionic Arm is the only additional device with a maximal load reported [68]. Its creators advised not to exceed a load superior to 5 kg [68].

### 3.4.3. Kinematic Specifications

*Range of motion.* Only the Kinetic Hand's creator mentioned information relative to some range of motion, its creator measured an 18° flexion angle for a full closure [82].

*Grasping.* Besides the Kwawu Arm 2.0—a prosthetic device for which we previously reported data concerning its possible grips—the "El Medallo" Bionic Arm is the only additional device provided with information relative to grasping ability [68]. Its creators mentioned that this device can perform power and pinch grips [68].

No creator shared data concerning any degrees of freedom of their prostheses [56,65–69,75–87]. No creator provided data concerning the number of joints of the 3DP prostheses [56,65–69,75–87].

### 3.4.4. Production Cost

No information related to the production cost was provided for any devices [56,65–69,75–87].

## 4. Discussion

This study attempted to access and assess all the open-source 3D-printed upper limb prosthetic devices available in the scientific literature and on online repositories up to date. Our findings, though limited, are encouraging and convey a view of the present situation on open-source upper limb 3D-printed prostheses. Indeed, only 6 articles, from 166 initial publications, corresponded to our selection criteria and 2 online databases, out of 17 searched repositories, hosting hundreds of 3D-printable designs, remained from our screening. To that, 12 additional devices were selected from the specialized e-NABLE platform. Our results showed that from an impressive amount of initial data, after applying a limited number of selection criteria, the final quantity of reliable information was limited.

### 4.1. Designs

We discuss the main characteristics pertaining to the successive stages from designing the digital model to its physical production.

### 4.1.1. Creation

All open-source prostheses, except the Cyborg Beast, included in the present study were designed by private individuals or communities. The Cyborg Beast, from which the Cyborg Beast 2 was designed by Zuniga et al. was the first open-source upper limb prosthetic device designed and produced by a scientific team and previously reported in the scientific literature [22,36]. Since its publication, we found no other open-source 3DP upper limb prosthetic devices developed by scientific teams for daily usage in the literature. All papers included here discussed the application of existing 3DP prostheses. Of course, some researchers were involved in initiatives such as e-NABLE, and they contributed to the design of the proposed devices [11]. Nevertheless, their contribution might not be known among most healthcare professionals (prosthetists, occupational therapist, orthopedists, etc.) working with possible recipients, as it was not reported in the literature. Moreover, it is noteworthy to indicate that reports of newly developed 3D-printed upper limb prostheses exist in the scientific literature but are not open-source [92]. Studies are accessible after subscription to publishing journals, and even though designing and 3D printing protocols might be then obtained, no access to printing files is granted. Open-source access to those creative groups' creations should be promoted in order to stimulate progress as it had been demonstrated in the open-source community for almost a decade. An exception to this common practice is the open-source 3DP The Handi Hand, developed by Brenneis et al. in 2017 for research purposes only [93].

It is also important to point out that most, if not all, available 3D-printable upper limb prostheses were designed by individuals in possession of their both healthy upper limbs, in order to help affected individuals. Therefore, a recipient-centered approach is crucial to ensure a fitting response to specific physical demands [1,11]. While many private makers (e.g., e-NABLE) are accustomed to working with affected recipients [18], others (e.g.,

online contests on Instructables) do not necessarily intend to answer specific needs [40]. Prosthetic devices uploaded in such context, or with unclear purposes, should thus be approached with precautions if considered appropriate to fit a potential recipient. The more designs meet specific requests, the more upper limb 3D-printed prostheses will evolve, as demonstrated by the improvement of multiple prosthetic devices on the e-NABLE community platform [18]. This is explained by the application of amputees' perception to the iteration process.

It is noteworthy that, as reported by Ten Kate et al [11], we also objectified that cosmetic-only upper limb prostheses are underrepresented. As cosmetic prosthetic devices remain the main choice for many recipients, more designs of such should be developed in order to meet their needs as much as those in need of functional prostheses [11,15].

As argued by Ten Kate et al. and Tanaka et al. [11,13], all professionals accustomed to designing, creating and fitting upper limb prostheses are necessary for a solid and reliable 3D-printed prosthetic device to be produced and ensure the implementation of that technology in this field for the benefit of potential recipients and their family.

### 4.1.2. Accessibility

It is essential to specify the nature of access to 3D-printed devices. Indeed, both commercial and free access exist with the associated monetary implications in terms of accessibility, production and maintenance for the users.

Moreover, a regular reassessment of 3DP prosthetic devices is mandatory as the review of Ten Kate et al. demonstrated that some devices and their printing files, or inspiring projects, might no longer be accessible, only a few years later [11]. Indeed, out of the fifty-one devices they found from online sources, eleven were no more accessible, with the majority being open-source prostheses, due to a non-functioning URL. Such inconveniences would be avoided by frequent reviews of available devices. The issue is similar with incomplete projects which lack parts (e.g., printing files, electronic code, wiring diagrams) that the creators promised to add in the future but to no avail.

### 4.1.3. Customization

The strength of 3D printing is the remarkable possibility of personalization of a design to obtain a tailored-made product. The level of personalization depends on the "maker's" experience and on the local resources such as design software, 3D printer quality and supplies. In the prosthetic field, one basic personalization step is to scale a 3D-printed device to match the recipient's affected upper limb size. Advanced personalization can involve different aspects of the design. Indeed, Tong et al. customized the socket of an original design after performing a 3D scan of a recipient's stump, which increased the uniformity of forces distribution along the limb [35]. Force distribution was secondarily analyzed through 3D-printed electrodes containing pressure sensors embedded in the customized prosthesis [35]. The study and experiments led by Tong et al. are to be highlighted. Indeed, 3D scanning is being more widely within the prosthetic field, as it allows precise morphological data collection, which leads to highly customized final devices, without requirements of advanced expertise [16,94]. Moreover, thanks to its democratization, 3D scanning is a relatively accessible technology [94]. Tong et al. proved that 3D scanning can also be incorporated in the open-source prosthetic field with positive results [35]. Furthermore, they also included electronic components in their open-source 3DP prosthesis which is also a valuable road to explore for this field [35]. Indeed, the use of embedded 3D-printed electronic components is also spreading [95]. This could open the door to additional functionalities for 3DP prostheses.

Additionally, the personalized passive adjustable developed by Alturkistani et al. allowed their participants to achieve professional tasks [34]. Numerous studies have shown that this aptitude is one of the main requests from adult prostheses users as often traumatic limb loss might lead them to cease their professional activities [1,15]. This positive reported result could only be achieved because the final recipient had been involved in the

development process. Likewise, Anderson et al. successfully customized an open-source 3DP prosthesis to meet the needs of a young athlete [33]. Nevertheless, one should be cautious about the expression of pain and discomfort from an infant. Indeed, as they fear a limitation of their activity, they might hide actual problems.

As a user-centered customization allows the production of a prosthetic device fitting meeting recipients' anatomy, forces are better distributed along the limb leading to a comfortable wear and potential lower abandonment rate [11,15,33–35]. Conversely, Omar et al. and Neethan al analyzed some prostheses' functionalities (e.g., grasping, grip strength) without resorting to amputated individuals for reliable measures [31,32]. Therefore, biases might have likely occurred, and results from their studies must be cautiously interpreted.

We arbitrarily included anthropomorphic upper limb 3DP prostheses only. Nevertheless, in view of the high abandonment rate of commercial anthropomorphic prostheses by users as demonstrated by Biddiss et al. [15], it is relevant to question, with potential future recipients, the actual need of such a prosthesis. A non-anthropomorphic terminal device (e.g., hook, universal tool holder) could be more appropriate depending on the daily life activities of recipients than a device mimicking a hand. However, it is also noteworthy to indicate that anthropomorphic devices can have significant positive psychological impact on impaired individuals, especially for younger patients [3,11].

Though not included in the scope of this study, 3D-printed finger prostheses should benefit from thorough assessment as finger amputations constitute one main cause of upper limb impairment [11]. However, such types of 3DP prosthetic device are under-represented, as also reported by Ten Kate et al. [11]. Hand prosthetic devices, followed by forearm prostheses, constitute the main designs available on online repositories.

### 4.1.4. Mechanical and Kinematic Specifications

As expected, mechanical and kinematic data were more detailed in scientific studies than on online repositories. However, those data, though crucial, were not reported systematically. As discussed throughout this discussion, creators and makers should particularly attempt to communicate information on the technical specifications of the printed devices for the benefits of recipients.

*Weight and maximal load*. Though the type of prosthesis and its actuation were commonly specified, data relative to weight and maximal load were often missing. This might be due to the relative unpredictability of those properties as they can considerably vary according to multiple factors such as scaling, printing settings (e.g., resolution, infill), printing material, and 3D printers' properties. Similarly, the durability of devices being a property influenced by those same factors as well as the constraints imposed and their usage (i.e., activities, duration, maintenance), it cannot be defined easily in advance [11,16,17]. This might explain the poor report of information on prostheses durability.

Out of all sources, only very few devices (e.g., Talon Hand, "El Medallo" Bionic Arm, NIOP Kwawu remix, Hackberry hand) [31,50,59,68] were either reported, or uploaded, associated with details concerning their weight or their maximal charge. The lack of this information can limit the capacity of a maker and a recipient to make an informed choice regarding a 3DP prosthesis. Indeed, they will impact the recipient's comfort (leverage, fatigue), participation in daily, and even possibly professional, activities and the risk of breakage. Nevertheless, some creators attempted to produce such data. Indeed, the maker "Profbink" disclosed the maximal charge the "Talon hand" prosthesis could lift (13 lb, ±6 kg) and provided an experimental video to support that information [50]. Such action could be reproduced by other private creators and would provide some data for potential users. Those data are to be considered carefully as printing settings do have an impact on the final device and its mechanical properties.

Therefore, reports of prosthetic devices including data relative to weight, strength or resistance properties should be accompanied by information on designing and printing settings. Such additional details would allow potential makers and users to consider a candidate prosthesis within the limits of their own skills and resources. Both scientists and private makers should be encouraged to share such information, though difficult to obtain, with their creations.

*Grasping and Range of motion.* Kinematic specifications inform the future recipient about the activities of daily living (ADL) achievable by acquiring a specific prosthetic device [11]. Therefore, this aspect is critical as it will influence the reinsertion of an amputee back into a social, even a professional life. Kinematic specifications are thus also mandatory as they will guide future recipients in their informed choice. As expected, very few prostheses from online databases were provided with such data, and remarkably, reports from scientific teams also lack important information. There is a distinct improvement required on that matter.

Grasping capabilities determine the future application and usage of a prosthesis. We can virtually expect that the more grasps a prosthesis can perform, the more it will be chosen and useful. Nevertheless, 3D printing technology and materials include physical constraints (e.g., rigidity, force distribution, actuation), so a typical prosthesis would perform two to four efficient grips [11]. Most devices would usually achieve the power and spherical grips, and some would also perform lateral and tip grips [11,38]. Of course, other grips can be achieved by open-source 3DP prostheses, depending on a device's configuration and functionalities.

Unfortunately, for most open-source 3D-printable prostheses, all their potential grips are rarely clearly specified before printing and testing the device, which can cause a mismatch between actual 3D-printed prostheses and recipients' needs. Nevertheless, practical expertise helps to anticipate the grasping abilities.

Scientific studies are also mandatory to subject press advertisement to reality [11]. Indeed, some seemingly revolutionary devices can happen to be of poor functional aid. For example, the Hackberry hand, though presented as an exceptional bionic upper limb prosthesis, demonstrated limited efficacity [96]. Indeed, Omar et al. studied the Hackberry prosthesis and concluded to its limited functionality due, in part, to a surface too rigid and slippery that hinders recipients' ability to perform efficient and sure grasps as well as some activities of daily life [31]. Nevertheless, the authors still considered the Hackberry as a low-cost (GBP 337) functional 3DP prosthesis [31]. Apparent technologically advanced devices can disappoint makers and recipients if not provided with sufficient evidences of grasping abilities and sure grip.

*3D printing.* As our results illustrated, most prostheses were 3D printed using the fused deposition modelling (FDM) technology with two main thermoplastic materials, ABS and PLA. This can be explained by its reported ease of use, limited cost and its capacity of fast production [16,17]. Nevertheless, with these seemingly appealing features, it is also important to understand some crucial printing parameters that have a direct impact on the tensile, flexural and impact strength of printed parts [97,98]. Among these printing settings, we note raster orientation, the printing temperature, the layer thickness and height [97]. Moreover, some factors are also known to cause weak printed parts with FDM, such as the porosity and high volume of air gaps leading to inter and intra layers deformation, weakening the final product [17,97,98]. Therefore, printing solid parts requires manufacturers to consider multiple variables simultaneously. Moreover, most FDM 3D printers accept only thermoplastic materials in comparison to some other 3D printing techniques. Indeed, additional printing techniques exist (e.g., selective laser sintering (SLS), stereolithography (SLA), inkjet printing (IP)), all differing by their printing mechanism, the materials they can print, their printing resolution, their printing mechanism or their capacity to printing complex objects [17]. Similarly, various printing materials are available on the market (e.g., thermoplastics, resins, metals, ceramics), each offering the possibility of additional applications thanks to their respective printing and mechanical properties [17].

Nevertheless, their often-associated high cost or required expertise can explain their poor presence in the field of open-source 3DP upper limb protheses.

It is also noteworthy to indicate that although most 3D-printed prostheses are made of hard plastic, a functional prosthesis should incorporate some flexible material in the superficial layers to enhance the grasping abilities [11]. Indeed, to ensure a congruous and sure grip around an object held, a human hand combines the interaction of three essential elements: the compliancy of the skin and soft tissues, the muscles contraction and the rigidity from the skeleton of the fingers and palm [99]. Therefore, to improve the grasping capacity and grip strength of prosthetic devices, 3D-printed upper limb prostheses should be produced with a combination of rigid parts and flexible material [11]. Currently, an increasing number of 3D printers are capable to extrude multiple printing materials during the same printing process [11,17]. This facilitates the incorporation of different material in a single printed device, leading to the production of hand prostheses with compliant palm and fingers [11,17].

3D printing is a complex technology integrating multiple interrelated settings. Therefore, determining which factors to report is not simple. However, some specific printing parameters do have a direct impact on a prosthetic device printed in terms of resistance, strength and durability. Among those crucial settings, we decided to highlight the infill percentage and pattern, the raster orientation, or the printing orientation, the layer height and the material used [11,16,100].

Indeed, Akhoundi et al. specifically studied the effects of infill percentage and patterns on the strength of 3D-printed parts and demonstrated their direct consequence [100]. Indeed, their flexural and tensile moduli were consecutively affected by those two factors. As upper limb prosthetic devices undergo various constraints, it is essential to know which parameters to adjust to reach the adequate resistance [100]. Additionally, raster orientation is closely associated with infill settings. Even though the latter determine the strength of a 3D-printed object, the printing orientation does play an important role in determining the actual capacity to lift loads or resist constraints [100]. In fact, Akhoundi et al. demonstrated that the highest mechanical properties are obtained when the raster is aligned with the loading direction [100]. Therefore, reporting the printing orientation would contribute to producing resistant prosthetic devices able to lift relatively consequent loads. This attribute can allow a recipient to use a prosthesis in daily activities, which enhances self-reliance and social inclusion [100].

Layer height is also an important printing setting to consider as it directly impacts the quality of a printed object [11,17]. Indeed, a low layer height increases the smoothness of a surface and details of a print, but it lengthens the printing time as layers are deposited, cured or sintered according to the 3D printing technology chosen [17,101]. Conversely, larger layer thickness results in a poor resolution with a rough surface in a shorter printing duration [17,101]. Moreover, experimental studies have demonstrated the impact of layers height on the tensile and compressive strength of printed parts [97,102]. Every 3D printing technology is associated with a specific layer resolution range. Here, are the layer resolutions for the four most employed 3D printing techniques for 3DP prostheses: FDM 50–200 μm, SLS 80–250 μm, IP 5–200 μm and SLA around 10 μm [17,101]. Finer details can be obtained with SLS than FDM. IP and SLA technologies allow both the production of highly complex geometries and high layer resolution [11,17,101]. Therefore, based on the purpose of printed parts, the layer height must be adjusted carefully.

Finally, as mentioned above, each printing material possesses specific printing and mechanical properties, the former influencing logically the latter and the final 3D-printed objects. Therefore, to facilitate the reproducibility of an original device, creators should also report, or recommend, the printing material to be preferentially employed [16].

　　　　Having access to data relative to infill settings, layer height, printing orientation and material will guide potential future makers and recipients in producing a solid and reliable 3D-printed upper limb prosthesis.

### 4.2. Assembly

　　　　As prostheses are no longer assembled by certified prosthesis, even though studies and platforms encourage to seek for their advices [18], clear and reliable printing and assembly instructions become essential. As not all makers have IT abilities, guidance should be provided for individuals making their first steps in this field. Incomplete mounting guidelines can lead to errors during the customization or printing processes, or to the incapacity to assemble a functional prosthetic device. Therefore, reliable instructions are crucial.

　　　　Additionally, in order to be assembled some 3D-printed upper limb prostheses require hardware such as specifics screws, drills or even hammers. Yet, such crucial information sometimes lacks in the instructions provided with the device which can become a limitation for potential makers. Nevertheless, when existing, a support community would enlighten information seekers on such details [18,39]. Omission of such important information could be avoided by systematically reporting all essential details concerning the design, printing and assembling processes both in scientific papers and on open-source repositories.

　　　　It is also relevant to point out the diversity of ways to communicate findings on online databases. For example, many prostheses' creators shared information such as assembly instructions through video. This method allows a creator to actually assemble a device step by step while sharing practical advice for future makers and recipients. This way of communication can be more productive and helpful than instructions provided in a classic written format in the scientific literature, though also used by some creators online. Some other makers shared detailed pictures illustrating the mounting process of their devices. Those different methods should also be studied in order to determine the most effective one. We can expect that there is a corresponding reporting method for each maker as individuals differ.

### 4.3. Validation

　　　　As reported by Diment et al., no study ever proved the clinical efficacy of 3D-printed upper limb prostheses [10]. Data from currently reported studies lack methodological strength, impeding thorough statistical analysis to validate any impacts of those devices in individuals' life [10]. Moreover, large scale studies exploring, through questionnaires or functional tests, recipient's usage, comfort, complaints as well as devices' mechanical properties (e.g., durability, strength) and kinematic specifications (e.g., range of motion, efficient grips) are not available. They should be undertaken, without delay, to support the implementation of 3D-printed prostheses in the clinical routine [10–12]. The contribution of these future data would guide policy makers in determining the position of those devices in the medical field as they have potential of helping many individuals.

### 4.4. Systematic Review

　　　　Many assessment tools exist to assess the quality and the efficacy of classic commercial devices (e.g., Box and Block test, Southampton test) as well as surveys to determine recipients' degrees of satisfaction [103,104]. Some included studies implemented these tools to assess the impact of 3D-printed upper limb prostheses [33,34,36]. For example, Anderson et al. reported the use of a survey to assess the effect of their 3DP prosthesis on their young participant's confidence, satisfaction and participation in activities [33]. Nevertheless, no validated test or survey is yet available to address specifically the global effectiveness of 3D-printed upper limb prosthetic devices in a daily usage environment. As the technology spreads in the scientific and healthcare community, specialized assessment tools should be developed based on those already existing. They would support large cohort studies, which are currently lacking.

*4.5. Online Databases*

Online databases offering a free access to 3D-printed upper limb prostheses projects are one of the keys to success for 3DP prosthetic field. Indeed, they gather projects, allow modifications, support and promote exchanges of experience, all of that free of charge for many of them [39,40]. They are the favored way to share findings and projects for private makers as well as for some non-profit organizations (e.g., e-NABLE, Unlimbited) [18]. Unfortunately, it is not possible to identify the number of contributions directly posted by scientific teams with open-source access. One example is the Cyborg Beast, which was uploaded, on one hand, by Jorge Zuniga on the Thingiverse platform and, on the other hand, was published in a peer-reviewed article, both on an open-source basis [11,22,36]. Scientific groups should be strongly encouraged to contribute to those open-source databases accessible to all.

It is noteworthy to indicate that duplicates are common, between and within online databases. Indeed, creators tend to spread their creations on different platforms and individuals are encouraged to upload their personal reproduction of a model, called a "remix", even though no substantial improvement was made. That phenomenon virtually increased the amount of 3D-printed upper limb prostheses. Moreover, many of the online repositories searched did not show the numbers of findings requiring a manual count, which increases the risks of quantitative errors.

Furthermore, though free online repositories contribute to the open-source access to reliable prosthetic devices, they can also host prosthetic devices, sometimes with no specific distinction, posted for educational or contests purposes only. Those inconveniences can confuse a recipient facing such a plethora of devices among which some were never tested even though possibly technologically appealing. Appropriate labels, added either by the platform or the creator, on such devices would help possible recipients choosing between candidate prostheses. Of courses, all databases are not equal on that matter. Some (e.g., Thingiverse, Instructables) tend to be associated with sufficient descriptions by creators. For example, the mention "Still in progress" is applicable on Thingiverse. Conversely, other repositories (e.g., Pinshape, Youmagine) host many designs of 3DP prostheses with very few descriptive information.

The e-NABLE community is a worldwide initiative allowing both professionals working with prostheses and private makers to meet and share respective expertise [18]. Moreover, it represents a valuable aid for individuals in need of an upper limb prosthetic device as they are eligible for a device independently of their financial resources. Indeed, a volunteering maker will scale, customize if possible and required, 3D-print, assemble and fit the prosthetic for a recipient free of charge [18].

Thanks to their long expertise, the non-profit organization developed its own rating system for devices which measures five aspects: maturity, cost of materials, popularity, grip strength and difficulty of production. Therefore, potential makers and recipients have access to additional valuable information to contribute to their informed choice about their future 3DP prosthesis [18].

More platforms, such as the e-NABLE platform, should appear to promote the spread of 3DP prostheses among the scientific and healthcare community and allow discussions and collaborations between them and the private makers or specialized non-profit organizations.

*4.6. Technical Support*

It is relevant to point out a difference in treatments relative to technical support between online databases and the e-NABLE platform. Indeed, it appeared that for some devices, initially hosted on another platform (e.g., Thingiverse), there was no more technical aid provided on it after some time but on the e-NABLE platform alone (e.g., Cyborg Beast, Flexy Hand 2) [22,23]. This is problematic as not all potential users are aware of or willing to be part of the e-NABLE community. They should be able to benefit from the inputs and improvements made on a device originally hosted elsewhere. Moreover, the absence of active feedbacks on an original online database could convey a false impression of

abandonment of a device project whereas it had been continuously employed and improved by the e-NABLE community. A regular update on the original device's hosting platform on latest progress should be systematically promoted. For example, the maker "Profbink", who is active in the e-NABLE community with some personal creations (e.g., Talon Hand, Ody Hand and Osprey Hand), keeps their native Thingiverse information pages updated as well as the support forum while they are being shared in the e-NABLE community [75,76,81].

### 4.7. Open-Source Era

The open-source era changed the paradigm of production of and access to medical products (e.g., devices, software) and therefore the accessibility of prosthetic devices. Open-source licensed devices can be accessed, acquired and modified by individuals different from their creators, leading to progress. Indeed, open-source licenses can offer the possibility to any individuals to customize and improve old devices for the benefits of potential recipients. For example, the Talon Ratchet Hand was achieved from the original Talon Hand with an add-on allowing the user to keep the hand closed with fingers flexed [105]. Numerous studies have reported that painful, even harmed, or strained stumps do contribute to the high abandonment rate observed among prostheses users [1,15,106]. Such improvement relieves pressure and constraints on the stump leading to a more comfortable experience for the user. Another example is the e-NABLE Osprey hand palm customizations designs, which are prosthetic hands derived from the Osprey hand, with its properties maintained, adjusted for recipients who would have some remaining functional fingers [87,107]. Such modifications allow individuals with a partial hand amputation and remaining fingers to use a prosthesis. This is relevant as finger amputations are prevalent, but the availability of adequate open-source 3DP prostheses is not high [1,11].

Even though the open-source access increases the accessibility to prosthetic devices and facilitates their development, it is also associated with some drawbacks. Indeed, it is noteworthy to underline the lack of medical supervision [11–13]. In order to 3D-print a prosthesis, one must consider recipient's specific anatomical characteristics to ensure a proper fit of the device [3,11,13,14]. Appropriate limb measurements, correct matching to the residual limb anatomy (e.g., remaining fingers, fingers shape, stump shape, stump homogeneity and stump size) and comfortable fitting are parameters considered by prosthetists to create a personalized commercial prosthesis [108]. Moreover, they also follow up their patients to prevent potential somatic damages [13,108]. Those vital skills to guarantee the production and the safe wear of a prosthesis by its users are not mastered by most prostheses' designers and makers [11,108]. This represents a main limit to open-source 3D-printed prostheses. Among other limitations of open-source prostheses, there is the unpredictability of mechanical properties [11,16]. Indeed, though makers would manipulate the same 3D prosthetic designs, their respective 3D printers and materials brands would likely differ [11]. Consequently, the 3D-printed prostheses would likely also present different mechanical and kinematic specifications. The reproducibility is therefore limited. Additionally, the volume of 3D printers can act as a limiting factor. Indeed, if the dimensions of a 3DP prosthesis are superior to those from a 3D printer, the 3D printing would not occur [11]. A potential recipient might either need to find another maker owning a 3D printer with a larger printing volume, or the prosthesis could be 3D-printed in multiple parts. In the first scenario, such search is not guaranteed to succeed depending on the recipient's place of residence, whereas in the second option, the mechanical properties would change with a possible impact of the prosthetic device strength and durability. Reporting printing settings, assembly and fitting details can help designers and makers to hinder the impact of those limitations.

A real, transparent and sincere open-source philosophy would lead to technological improvements for the benefit of affected individuals. To facilitate those developments, open-source licenses should become mandatory for 3D-printed upper limb prostheses in order to assure their accessibility to all. Nevertheless, the limitations due to the open access

model should also be actively addressed to ensure the most adequate and fitting prostheses to all potential users.

### 4.8. Limitations

Our work is not without limitations. Indeed, attempting to circumscribe a field which does not possess specific guidelines brings both advantages and limits. Therefore, our selection criteria can be discussed.

Firstly, we arbitrarily set 2018 as the threshold year as development can both evolve and become extinct as rapidly as our review of the devices reported by Ten Kate et al. highlighted [11]. As exemplified by our results, some reliable upper limb prostheses were developed before 2018, so there is a clear risk of omitting some of them, such as the Talon Hand 3.0 or the Ody Hand [75,76]. Nevertheless, we did strive to reduce our blind spots by following an overinclusive methodology by searching numerous online repositories and by including older devices associated with technical support up to this day. Additionally, our search of multiple online databases led to duplicates which lowers the chance of missing out reliable devices. Therefore, the date of last updates only cannot be employed alone as a constraining excluding factor but associated with the absence of any proof of updates on the device can help the potential user determine its reliability and the support possibly available. For example, the e-NABLE organization still promotes the development and use of the upper limb prostheses available since before 2018. Their up-to-date repository is also a valuable resource in determining the reliability of older devices [18,19].

Secondarily, our restrictive criteria concerning the provision of sufficient and reliable resources to potential users for comfortable printing and assembly processes can also be reasonably discussed as it can appear to be a limiting factor. Indeed, many of the open-source upper limb prosthetic devices available online were designed, conceived and offered by volunteer private citizens who often desire to help family members or relatives [39,40,75]. Those "makers" are not all professional designers, engineers, hand therapists, prosthetists or healthcare providers. Therefore, they might not follow the same codes and requirements to report their findings and works. Thus, expecting strict reports to validate a device could hinder the motivation and involvement of a whole volunteering worldwide network. Guidelines to ensure a reliable report of data should be developed for makers.

Imposing a specific language to report data and experiments can be inappropriate as first contributors are citizens who communicate in their native language. Restricting the communication codes could have a negative impact on the participation of willing people in this global effort. Contrary to scientific literature, multi-language contributions should be promoted, which would benefit open-source progress of the technology as every contributor would share findings without language barriers.

Thirdly, consecutively to our selection criteria, we excluded scientific articles that did not mention the use or the potential availability of their 3D-printed designs. Unfortunately, that approach possibly also excluded devices that could have been accessed by contacting studies' authors. If so, undertaking such action can be common for researchers whereas it can appear to be very intimidating for private citizen. Accessibility of prosthetic devices designs should be clearly stated in scientific papers.

Fourthly, our selection criterion consisting in including only open-source, or freely accessible, devices could be considered as reductive. Indeed, there are numerous 3D-printed upper limb prosthetic devices reported in the scientific literature, but only a very narrow niche consists of open-source or freely available devices as demonstrated. Nevertheless, that limited access to potential life-changing technology might be controversial. Indeed, the population in need of prosthetic devices who opt for open-source 3D-printed options could be basically classified into two categories:

1° Individuals living in western countries without any health insurance, and incapable to afford prostheses for themselves or their infants;

2° Individuals living in third-world countries or in a (post-) war environment. They have a limited, if not absent, access to healthcare. They often have no regular income.

Facing such groups, scientific groups should opt for sharing their devices, or at least their newly acquired technology, openly, in order to help the community. Having 3D-printed upper limb prosthetic devices with their associated technology inaccessible due to a financial barrier would slow down volunteers' effort.

An exchange of knowledge and experience between scientists and makers should be actively promoted which would contribute to refine existing designs and lead to more efficient and accessible ones.

Despite our theoretical inclusion criteria, unreliable, or non-functional, protheses might mistakenly be included in our study, as if considered reliable. Therefore, a systematic, solid and repetitive testing protocol by a qualified multidisciplinary team (e.g., orthopedists, prosthetists, occupational therapists) should be regarded as the best assessment procedure for a prosthetic device to be approved. Conversely, even though we attempted to broaden our research, we likely might have omitted few designs.

*4.9. Future Perspectives*

Our results illustrated that the research setting available to scientific groups and the expertise from private individuals or specialized communities can and should be complementary. More 3D-printable upper limb prosthetic devices developed by scientific teams should be accessible freely online, or under an open-source license, in order for private initiatives to benefit from their technological advancements. In parallel, scientific groups should study reliable open-source 3D-printed prostheses already available to provide technical data relating to their characteristics and usage, essential for their improvement. Promoting exchanges between private makers (e.g., e-NABLE community) and professionals (prosthetists, orthopedists, occupational therapists, etc.) through unanimously recognized specialized platforms (i.e., approved and validated forums or journals) dedicated for that purpose should be the rule. A balance between usual scientific writing rigor and simple and creative communication, though solid and structured, must be found. As mentioned above, some private groups developed an expertise worthy to be considered by confirmed scientists even though communication standards differ. An open, global and consistent dialogue between all partners worldwide is to be begun. Learned societies of related medical specialties (e.g., orthopedists) can play a vital role in that field.

Most of the devices available are provided without, or with limited, data concerning their mechanical, kinematic and technical characteristics as well as the printing and assembly processes. Such information is crucial to aid potential makers and recipients in making an informed choice about a 3DP prosthesis among the diversity of prosthetic devices available. In that instance, we suggest a checklist with some essential information to be reported when uploading or presenting a device. That checklist was based on our search and the lacking data in reports. Information requiring objective measures (e.g., range of motion) should be supported by brief but solid evidence (e.g., measurements, experimental video). Data (e.g., measurements) not acquired should be clearly stated. See Table 7 for our 3D-Printed Prosthesis Report Checklist (3DPRC) and Appendix A for examples of applications of that checklist. Creators of 3D-printed prosthesis hosted on online repositories and specialized community platforms (e.g., e-NABLE) should ensure that any updates on their device designs and their associated files (i.e., printing files, instructions, etc.) are accessible on all platforms on which they chose to share them.

Regulations should not be implemented to circumscribe this field, as it would progressively hinder contributions from volunteering private individuals. Instead, guidelines should be developed by experts (e.g., makers, physicians, NGOs leaders, recipients) to ensure some level of consistency, while allowing creativity in designing, producing and fitting 3D-printed upper limb prostheses for individuals in need.

**Table 7.** 3D-Printed Prosthesis Report Checklist.

| Information | Reported |
| --- | --- |
| Identification | |
| • Creator | |
| • Year of Creation | |
| • Number of versions | |
| 3D printing | |
| • Design customization required | |
| • Material (type) | |
| • 3D printer | |
| • Orientation | |
| • Infill (percentage, pattern) | |
| • Printing recommendations | |
| • Printing duration | |
| • Post-printing process | |
| • Cost | |
| Mechanical specifications | |
| • Level of prosthesis | |
| • Type of prosthesis (passive, active) | |
| • Actuation (body-powered, externally powered, etc.) | |
| • Type of control (wrist, elbow, shoulder harness, EMG, EEG, etc.) | |
| • Weight of device (if applicable) | |
| • Maximal Load (tests performed) (if applicable) | |
| • Durability (if applicable) | |
| Kinematic specifications | |
| • Grasping (according to GRASP taxonomy) | |
| • Range of motion (in degrees) (if applicable) | |
| • Degree of freedom (if applicable) | |
| • Force distribution (if applicable) | |
| Assembly | |
| • Material (hardware required, cost) | |
| • Assembly recommendations | |
| • Assembly duration (if applicable) | |
| Application (if applicable) | |
| • Population (age, impairment, usage, activities) | |
| • Intervention (test, procedures, etc.) | |
| • Presence of control group | |
| • Main results | |

Finally, field data about 3DP prostheses are lacking. Therefore, real-life surveys of individuals employing 3D-printed prostheses are greatly expected and mandatory. Such studies would investigate information such as the resistance of devices, need of reprints, the comfort and limits of use, and the abandonment rate. In addition, randomized clinical trials in this field should also be considered a priority to demonstrate the clinical efficacy of upper limb 3D-printed prostheses.

**5. Conclusions**

Open-source 3D-printed upper limb prostheses stand as potential alternatives to classic commercial prosthetic devices. Many individuals worldwide opted for them as they represent functional and affordable solutions. Nevertheless, much research is warranted to improve their robustness and objectively assess their impact on recipients' daily life. Clear and validated guidelines are required for indications, production and fitting in order to expedite the implementation of 3D-printed prostheses in clinical routine and their acceptance by healthcare providers community (orthopedists, hand therapists, etc.).

**Author Contributions:** Conceptualization, K.W., O.B., B.R. and R.O.; data curation, K.W. and R.O.; writing—original draft preparation, K.W. and R.O.; writing—review and editing, K.W., X.B., T.S. and R.O.; supervision, T.L., B.R. and R.O. All authors have read and agreed to the published version of the manuscript.

**Funding:** Oral and maxillofacial surgery Lab, NMSK, IREC, UCLouvain, Brussels, Belgium.

**Institutional Review Board Statement:** Not applicable.

**Informed Consent Statement:** Not applicable.

**Data Availability Statement:** The data used to support the findings of this study are available from the corresponding author upon request.

**Conflicts of Interest:** The authors declare no conflict of interest.

## Appendix A  Examples of Applications of the 3D-Printed Prosthesis Report Checklist

Example n°1: Zuniga et al. [36].
Example n°2: Omar et al. [31].
Example n°3: Kinetic Hand [79].

| Example n°1. 3D-Printed Prosthesis Report Checklist | | Zuniga et al. [36] |
|---|---|---|
| **Information** | **Reported** | **Data** |
| Identification | | |
| • Creator | | Zuniga et al. |
| • Year of Creation | | 2017 |
| • Number of versions | | 2 |
| • Availability (open-source, URL) | | Open-source |
| 3D printing | | |
| • Design customization required | | Yes (colors, fictional characters) |
| • Material (type) | | ABS, PLA |
| • 3D printer | | Ultimaker 2, Ultimaker B.V., Geldermalsen, The Netherlands; Uprint SE Plus by Stratasys, MN |
| • Orientation | | Hexagon pattern (desktop printer), crosshatch (industrial printer) |
| • Infill (percentage, pattern) | | 40% |
| • Layer height | | 0.15–0.25 mm |
| • Printing recommendations | | Rafts and supports for delicate parts |
| • Printing duration | | 4–7 h (assembly time included) |
| • Post-printing process | 🟥 | |
| Mechanical specifications | | |
| • Level of prosthesis | | Hand |
| • Type of prosthesis (passive, active) | | Active |
| • Actuation (body-powered, externally powered, etc.) | | Body-powered |
| • Type of control (wrist, elbow, shoulder harness, EMG, EEG, etc.) | | Wrist |
| • Weight of device (if applicable) | 🟥 | |
| • Maximal Load (tests performed) (if applicable) | 🟥 | |
| • Durability (if applicable) | 🟥 | |
| Kinematic specifications | | |
| • Grasping (according to GRASP taxonomy) | | Cylindrical and Tip grasps |
| • Range of motion (in degrees) (if applicable) | | 20°–30° wrist flexion for hand closure |
| • Degree of freedom (if applicable) | 🟥 | |
| • Force distribution (if applicable) | 🟥 | |

| Assembly | | |
|---|---|---|
| • Material (hardware required, cost) | 🟩 | Nylon cord, elastic cord, Velcro, medical-grade firm padded foam, protective skin sock, BOA dial tensioner |
| • Assembly recommendations | 🟥 | |
| • Assembly duration (if applicable) | 🟩 | 4–7 h (printing time included) |
| Application (if applicable) | | |
| • Population (age, impairment, usage, activities) | 🟩 | 11 children (3–15 years old). Congenital defect or amputation. |
| • Intervention (test, procedures, etc.) | 🟩 | Function testing (Box and Block Test) Strength measurements (strength testing with dynamometer) |
| • Presence of control group | 🟩 | Non |
| • Main results | 🟩 | Improvement of manual gross dexterity (function); no significant impact on strength of residual wrist. 3DP prosthesis can be used as a transitional device to improve function. |
| Remarks : Reported and absent data are represented by green and red marks, respectively. | 🟩 | |
| **Example n°2. 3D-Printed Prosthesis Report Checklist** | | **Omar et al. [31]** |
| **Information** | **Reported** | **Data** |
| Identification | | |
| • Creator | 🟩 | Omar et al. |
| • Year of Creation | 🟩 | 2019 |
| • Number of versions | 🟥 | |
| • Availability (open-source, URL) | 🟩 | Open-source |
| 3D printing | | |
| • Design customization required | 🟥 | |
| • Material (type) | 🟩 | PLA |
| • 3D printer | 🟥 | |
| • Orientation | 🟥 | |
| • Infill (percentage, pattern) | 🟥 | |
| • Layer height | 🟥 | |
| • Printing recommendations | 🟥 | |
| • Printing duration | 🟥 | |
| • Post-printing process | 🟥 | |
| • Cost | 🟩 | PLA filament: £37 Hardware parts: £300 3D printer: £310 |
| Mechanical specifications | | |
| • Level of prosthesis | 🟩 | Hand |
| • Type of prosthesis (passive, active) | 🟩 | Active |
| • Actuation (body-powered, externally powered, etc.) | 🟩 | Externally powered |
| • Type of control (wrist, elbow, shoulder harness, EMG, EEG, etc.) | 🟩 | Infra-red sensor |
| • Actuators | 🟩 | Servo motors |
| • Weight of device (if applicable) | 🟥 | |
| • Maximal Load (tests performed) (if applicable) | 🟩 | 2 kg |
| • Durability (if applicable) | 🟥 | |
| Kinematic specifications | | |
| • Grasping (according to GRASP taxonomy) | 🟩 | Power, Tip, Lateral and Spherical |
| • Range of motion (in degrees) (if applicable) | 🟥 | |
| • Degree of freedom (if applicable) | 🟥 | |
| • Force distribution (if applicable) | 🟥 | |

| Assembly | | |
|---|---|---|
| • Material (hardware required, cost) | 🟩 | PCB board, infra-red sensors, servo motors |
| • Assembly recommendations | 🟥 | |
| • Assembly duration (if applicable) | 🟥 | |
| Application (if applicable) | | |
| • Population (age, impairment, usage, activities) | 🟥 | |
| • Intervention (test, procedures, etc.) | 🟩 | Testing (achievement of ADL tasks) |
| • Presence of control group | 🟥 | |
| • Main results | 🟩 | Limited hand functionality (limited grasps options) |
| Remarks: Reported and absent data are represented by green and red marks, respectively. | 🟩 | |
| **Example n°3. 3D-Printed Prosthesis Report Checklist** | | **Kinetic Hand** |
| **Information** | **Reported** | **Data** |
| Identification | | |
| • Creator | 🟩 | Mat Bowtell (Free 3D Hands, Ltd.) |
| • Year of Creation | 🟩 | 2020 |
| • Number of versions | 🟩 | 1 |
| • Availability (open-source, URL) | 🟩 | Open-source ([https://www.thingiverse.com/thing:4618922](https://www.thingiverse.com/thing:4618922), accessed on 3 May 2022) |
| 3D printing | | |
| • Design customization required | 🟥 | |
| • Material (type) | 🟩 | PLA, PLA+, Ninjaflex |
| • 3D printer | 🟩 | Flashforge Finder Lite, Flashforge Creator Pro |
| • Orientation | 🟥 | |
| • Infill (percentage, pattern) | 🟩 | 40%(PLA)/100% (Ninjaflex) |
| • Layer height | 🟩 | 0.18 mm |
| • Printing recommendations | 🟩 | No support, no raft See complete instruction manual for full printing recommendations |
| • Printing duration | 🟥 | |
| • Post-printing process | 🟩 | See complete instruction manual for full post-printing recommendations. |
| • Cost | 🟥 | |
| Mechanical specifications | | |
| • Level of prosthesis | 🟩 | Hand |
| • Type of prosthesis (passive, active) | 🟩 | Active |
| • Actuation (body-powered, externally powered, etc.) | 🟩 | Body-powered |
| • Type of control (wrist, elbow, shoulder harness, EMG, EEG, etc.) | 🟩 | Wrist |
| • Actuators | 🟥 | |
| • Weight of device (if applicable) | 🟥 | |
| • Maximal Load (tests performed) (if applicable) | 🟥 | |
| • Durability (if applicable) | 🟥 | |
| Kinematic specifications | | |
| • Grasping (according to GRASP taxonomy) | 🟥 | |
| • Range of motion (in degrees) (if applicable) | 🟩 | 18° wrist flexion for full closure |
| • Degree of freedom (if applicable) | 🟥 | |
| • Force distribution (if applicable) | 🟥 | |
| Assembly | | |
| • Material (hardware required, cost) | 🟩 | See complete instruction manual for full hardware lists. |

| | | |
|---|---|---|
| ● Assembly recommendations | <span style="background-color:green"> </span> | See complete instruction manual for full assembly procedure. |
| ● Assembly duration (if applicable) | <span style="background-color:red"> </span> | |
| Application (if applicable) | | |
| ● Population (age, impairment, usage, activities) | <span style="background-color:red"> </span> | |
| ● Intervention (test, procedures, etc.) | <span style="background-color:red"> </span> | |
| ● Presence of control group | <span style="background-color:red"> </span> | |
| ● Main results | <span style="background-color:red"> </span> | |
| Remarks: Reported and absent data are represented by green and red marks, respectively. | <span style="background-color:red"> </span> | |

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
