# Peer review of "Open-Source 3D Printing in the Prosthetic Field—The Case of Upper Limb Prostheses: A Review"

_machines, doi:10.3390/machines10060413_

Round 1

Reviewer 1 Report

It is difficult to follow the article in its current form

Schematic and images are necessary to provide the flow of the writing. I can not provide a clear decision in the current form. Authors are recommended to provide more detail with images and schematics to have a full review report

Reviewer 2 Report

Comments are attached 

Reviewer 3 Report

You write (line 102): "We determined criteria to virtually assess the accessibility and reliability of those prostheses as no specific protocols exist." Further, you present (Table 7) "the checklist with some essential information to be reported when uploading or presenting a device". Why you did not fill in some data in Table 7 to show the effects of your research? In my opinion, the basic condition for comparing 3D printed prostheses should be the opinion of orthopedists. Only on the basis of this review, we cannot judge the usefulness of the presented prostheses.

Individuals living in western countries without any health insurance or individuals living in third-world countries or in a (post-) war environment  (Line 911) will not be able to use 3D printing by themselves and some charities will have to do this, therefore only models that have been practically tested and approved by orthopedists will be allowed.

The search strategy presented in APPENDIX A is very simple, if there is any strategy at all. There are repetitions for search queries.  This Appendix could be canceled.

Round 2

Reviewer 2 Report

The revision is satisfactory and can be accepted in current form. 

Author Response

Thank you. We appreciated your relevant comments.

This manuscript is a resubmission of an earlier submission. The following is a list of the peer review reports and author responses from that submission.